# Real-time terahertz imaging with a single-pixel detector

Rayko Ivanov Stantchev [1✉], Xiao Yu[2,3], Thierry Blu [1] & Emma Pickwell-MacPherson[1,3]

Terahertz (THz) radiation is poised to have an essential role in many imaging applications, from industrial inspections to medical diagnosis. However, commercialization is prevented by impractical and expensive THz instrumentation. Single-pixel cameras have emerged as alternatives to multi-pixel cameras due to reduced costs and superior durability. Here, by optimizing the modulation geometry and post-processing algorithms, we demonstrate the acquisition of a THz-video (32 × 32 pixels at 6 frames-per-second), shown in real-time, using a single-pixel fiber-coupled photoconductive THz detector. A laser diode with a digital micromirror device shining visible light onto silicon acts as the spatial THz modulator. We mathematically account for the temporal response of the system, reduce noise with a lock-in free carrier-wave modulation and realize quick, noise-robust image undersampling. Since our modifications do not impose intricate manufacturing, require long post-processing, nor sacrifice the time-resolving capabilities of THz-spectrometers, their greatest asset, this work has the potential to serve as a foundation for all future single-pixel THz imaging systems.

[1] Chinese University of Hong Kong, Electronic Engineering, Hong Kong SAR, China. [2] State Key Laboratory of Power Transmission Equipment & System Security and New Technology, School of Electrical Engineering, Chongqing University, Chongqing 400044, China. [3] University of Warwick, Physics, Warwick CV4 7AL, UK. ✉email: rayko@cuhk.edu.hk

The information obtained from imaging with THz radiation has generated a great body of research showing potential applications in very diverse areas, see refs. [1–4] and references therein. Non-invasive THz inspections of semiconductor wafers[5] have yielded insight into the photocarrier dynamics of organic solar cells[6], observed crystal quality degradation in perovskites under UV illumination[7], and measured the conductivity of graphene[8] and indium-tin-oxide[9] samples. Further, the THz transparency of many common materials has been exploited to reveal sub-subsurface defects in carbon-fiber reinforced composites[10] and sub-structures of paintings[11]. In addition, clinical use of THz imaging is a highly sought after goal as the high-water sensitivity of these non-ionizing waves gives a label-free imaging contrast between healthy and cancerous tissue[12] and other afflictions[13].

THz-imaging technology, despite showing great application potential, is yet to be employed in the real-world most notably due to high equipment costs and/or needing highly trained operators. The reason for this is, unlike the visible domain where there are megapixel imaging arrays in all modern smartphones, the construction of a THz pixelated-detector array is problematic simply due to a lack of suitable materials; thus most current multipixel THz-detector arrays are narrowband or require cryogenic temperatures[14,15], although microbolometer arrays offer great potential as they can detect large bandwidths at room temperature[16]. In fact, microbolometer arrays have been used to obtain amplitude and phase images at a single THz frequency via the use of digital holography[17,18]. However, spectral images would require taking multiple images at different frequencies and would not achieve the sub-picosecond temporal resolution granted by THz-spectrometers, a capacity needed for observing ultra-fast dynamics. Although one can use electro-optic THz detection combined with an optical CCD array to map out a THz image[19] and circumvent the above problems, this requires a low-repetition rate regenerative amplified Ti:Sapphire laser rendering the entire system expensive, bulky and inflexible.

Recent advances in spatial light modulators have opened the door to alternative imaging technologies, notably the use of single-element detectors in conjunction with spatial encoding masks[20–24]. In this imaging modality, one encodes a beam of light with a time-varying spatial mask and records the transmission (or reflection) from an object with a single-element detector. Then knowledge of the spatial patterns and the detector readouts are combined to reconstruct an image of the object[20,25]. Not only THz single-pixel cameras have gathered recent attention[26–28] as they offer a viable path towards cheap and robust, no-moving-parts THz-imaging systems, thus eliminating the need for linear $xy$-stages used in other interference-based THz-imaging setups such as reference[29], but also because they have been used to study various systems at sub-wavelength resolutions[30,31]. Namely, a single-pixel THz camera that was used to perform near-field studies of graphene[32], revealed micron-sized circuitry defects hidden by silicon[33] and even made it possible to study sub-wavelength sized biological matter[34]. However, these works have all had slow imaging speeds: there is yet to be an explicit investigation into the modulation depth, switch-speeds and software design of such THz single-pixel imaging systems.

Here, we consider and optimize the factors that affect the efficiency of operation in this THz-imaging scheme. We experimentally show the most efficient geometry for the spatial modulator to be a total internal reflection (TIR) configuration. Then, by considering the temporal response of the system, we show that with post-processing one can reach a mask modulation rate of 20 kHz, only limited by the visible-light spatial modulator where otherwise physical phenomena limit us to 6–8 kHz. We further optimize the system by applying a carrier-wave modulation and

adjusting the equations for further gains in signal-to-noise without needing a lock-in amplifier. To conclude, we deliberate image undersampling methodologies with fast image reconstruction to reduce the total acquisition time, culminating to the acquisition of a THz video with $32 \times 32$ resolution and 6 frames per second displayed in real-time using a single-pixel detector. This significantly improves up on the current state-of-the-art which achieved $8 \times 8$ resolution at 1 frames per second[26].

## Results

**Imaging with a spatial THz modulator.** To obtain an image with a spatial modulator and some single-pixel detector the following is performed: a beam of radiation is patterned with a time-varying mask and the transmission (or reflection) is simultaneously recorded through an object with a single-element detector. Then knowledge of the temporal-variation of the mask is combined with the temporal-readout of the detector to reconstruct an image of object as follows; if our object has a spatial transmission function $\psi(x, y)$ and if at time $t$ our mask had a spatial response described by $a(t, x, y)$, then our detector readout at this time is $y(t) = \iint \psi(x, y)a(t, x, y)\mathrm{d}x\mathrm{d}y$. This idea is most elegantly expressed as a matrix equation

$$Y = A\Psi, \qquad (1)$$

where the $i$th entry of $Y$ is our $i$th detector readout, the $i$th row of $A$ is the spatial mask $a(t_i, x, y)$ reshaped into a 1D vector and $\Psi$ is the 1D vector version of our object's image. The image is obtained via matrix inversion methods[21,23,25,33], and is compatible with compressed sensing algorithms where there are fewer measurements than the number of pixels in the image[24,31,35]. These ideas are thoroughly discussed in section Undersampling, and the images in "Software Optimization" section were obtained with a fully sampled Paley type-I Hadamard matrix.

For the above idea to be implemented, spatial modulation of a beam of radiation is needed. The most common THz modulators employ conductivity changes in some material for the control of THz radiation. For example, in high resistivity silicon, the Drude plasma frequency is below the THz regime yielding a dielectric material response. However, upon intense optical excitation, the increase of charge carriers moves the Drude plasma frequency above the THz frequencies switching the material response to that of a poor-conductor[28,33,36]. Further, by spatially patterning the optical pump beam, one can create a spatial THz modulator as the illuminated regions are rendered conductive and the other regions are left transparent. Alternatively, electrical injection/depletion of charge carriers can also be used to this effect[37,38]. Although an electrical modulator would be preferable as it would not need an optical pump source, hence reducing the system size, and the switch rates would be limited by electronics as opposed to other physical processes (with appropriate design), an optical pump spatial THz modulator is yet easier to be implemented with current technology. In any case, the ideas and methods described in the rest of this work are equally applicable for both electrically and optically controlled THz modulators. However, readers need to be aware that we use an optically controlled THz modulator; high resistivity silicon (>2000 $\Omega$ cm, 500-µm-thick) illuminated by a 450 nm continuous-wave (CW) laser diode.

Note, in this work we use a commercial THz time-domain spectrometer (MENLO K15), which uses a pair of fiber-coupled photoconductive antennae as the emitter-receiver pair. An optical delay line changes the temporal arrival times of the femtosecond pulses incident onto the emitter and receiver allowing the measurement of the temporal THz waveform. A DAQ card (NI-USB 6351) reads out the detector and bias state of the emitter instead of the built-in Lock-In amplifier.

**Modulation geometry.** Our modulation relies on controlling the conductivity of a material, however the angle of incidence onto the controlled material is also important. Liu et al., derived the Fresnel coefficients for an infinitely thin conductive layer, conductivity $\sigma$, in-between two materials of refractive indices $n_{1,2}$[37,39]:

$$r_s = \frac{n_1 \cos\theta_i - n_2 \cos\theta_t - Z_0\sigma}{n_1 \cos\theta_i + n_2 \cos\theta_t + Z_0\sigma}, \quad (2a)$$

$$t_s = \frac{2n_1 \cos\theta_i}{n_1 \cos\theta_i + n_2 \cos\theta_t + Z_0\sigma}, \quad (2b)$$

where $Z_0$ is the free-space impedance (377 Ω) and $\theta_{i,t}$ are the incident and transmitted angles, respectively, related to each other via Snell's law. Here, we only show the s—polarization equations although the p—polarization takes a similar form[39]. It is obvious that the reflection case can go to zero more easily due to the subtraction of $Z_0\sigma$ in the numerator of Eq. (2a), whereas Eq. (2b) requires much larger changes in $Z_0\sigma$ to make $t_s$ tend to zero. With this in mind, an experimental investigation into this phenomenon needs to be carried out.

We study three different modulation geometries: transmission, ordinary reflection and total internal reflection (TIR) as shown in Figs. 1a–c with the angles of incidence on the conductive layer being 8.5°, 8.5° and 30°, respectively. Figure 1d shows our results for the modulation depth, defined as $M = \frac{\sum_\omega |\mathscr{F}\{E_{\text{ref}}(t) - E_{\text{mod}}(t)\}|}{\sum_\omega |\mathscr{F}\{E_{\text{ref}}(t)\}|}$, where $E_{\text{ref,mod}}(t)$ is our reference and modulated THz fields (Supplementary Note 1) and $\mathscr{F}$ is the Fourier transform, for different excitation powers. Note that the same silicon wafer used in the reflection and transmission measurements was placed on top of the silicon prism for truthful comparison. Overall, as expected all modulation geometries achieve higher modulations with increasing pump power, however they all appear to be saturating at high pump powers (>500 mW cm$^{-2}$) with transmission, reflection and TIR reaching a maximum modulation of 0.77, 0.57 and 0.91, respectively. The discrepancy between modeling and experiment regarding the saturation behavior is discussed later.

TIR outperforms the other geometries with ordinary reflection being the least efficient for all pump powers. Although from Eqs. (2a) and (2b) one would expect transmission to be the worst, this is not the case because ordinary reflection suffers from the largest insertion losses. In particular, if we have a silicon wafer, as in Fig. 1b, then the first silicon interface will create a preliminary reflection that will be measured as can be seen in Supplementary Fig. 1. This preliminary reflection is not influenced by the photoexcitation of the top silicon interface, hence it appends a large unmodulated THz signal onto our detector. Although we can remove this preliminary reflection in the time-domain, doing so would be misleading in regards to the overall efficiency. Experimental removal of this unwanted pulse by using a thick silicon wafer that spatially separates these two pulses is possible, however this increases insertion losses. Another idea is to modulate this first reflection (i.e. photoexcite the bottom silicon interface in Fig. 1b), however one would be wasting all of the power transmitted into the silicon. The best solution to reducing these insertion losses is to simply change the angle of incidence so that the THz-wave is totally internally reflected at the photoexcited interface. For a silicon-to-air interface, the critical angle is 16.9°, which is not attainable simply by changing the angle of incidence onto a flat silicon wafer. Therefore, we use a silicon prism to achieve the desired angle of incidence as shown in Fig. 1c. This is why TIR performs best: it has the same insertion losses as transmission while achieving a larger modulation depth as predicted by Eq. (2). Although these equations provide an

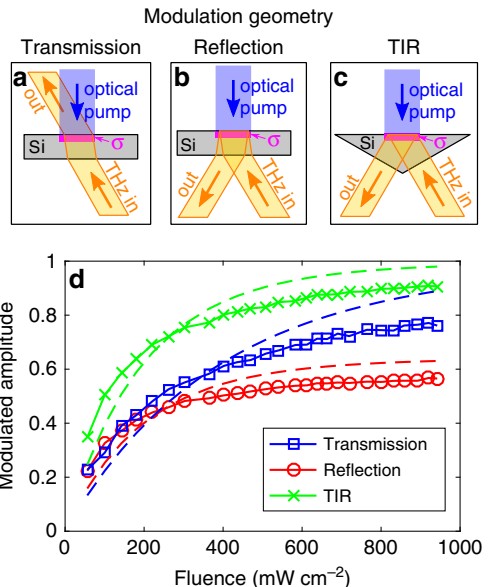

**Fig. 1 Modulation depth versus excitation power. a–c** The investigated modulation geometries; transmission, reflection and total internal reflection (TIR) respectively. A CW optical pump (450 nm) photo-excites a conductive layer (indicated by $\sigma$) in silicon. A THz beam is launched at the conductive layer and a detector collects the radiation in the direction indicated by "out" in each geometry. The incident angles of the THz beam onto the conductive layer are 8.5°, 8.5° and 30°, respectively. Note that the same silicon wafer used in **a** and **b** was placed onto top of the silicon prism in **c** for truthful comparison. **d** THz modulation depth as a function of pump fluence for the different geometries in **a–c** with the dashed lines being modeling.

intuitive explanation of the observed phenomenon, they do not provide a complete description of our phenomenon.

We photoexcite charge carriers in silicon with a CW source, and these carriers will diffuse inside the semiconductor with a characteristic length around ~0.3 mm rendering the infinitely thin conductive layer approximation invalid. Therefore, we split our silicon into a multi-layer structure[40], calculate the carrier concentration in each layer taking into account carrier diffusion and then use the Drude-model to calculate the permittivity of each layer (see Supplementary Note 2 for mathematical details). Although we have accounted for most of the relevant physics, we have not taken into account charge-carrier saturation mechanisms such as carrier-carrier screening and increased carrier collision rates at high concentration densities[41]. This is why modeling predicts higher modulation depths at higher pump powers than observation. A detailed discussion of these photo-carrier saturation phenomena would detract from the main aim of this work, namely optimize single-pixel THz-imaging methodologies. Therefore here it is sufficient to acknowledge that they will exacerbate further gains in modulation depth by increasing the pump fluence.

Knowing that the highest modulation depths are achieved in a TIR geometry, we construct our single-pixel THz camera accordingly as shown in Fig. 2; a collimated THz beam passes through an object and an image of the object is projected on the top interface of a silicon prism. Simultaneously we project, via a digital micromirror device (DMD), a spatially patterned 450 nm laser light beam on this silicon interface. As discussed previously, this imparts the spatial pattern in the 450 nm beam to the THz beam, and finally the THz is detected by a single-pixel photoconductive antenna.

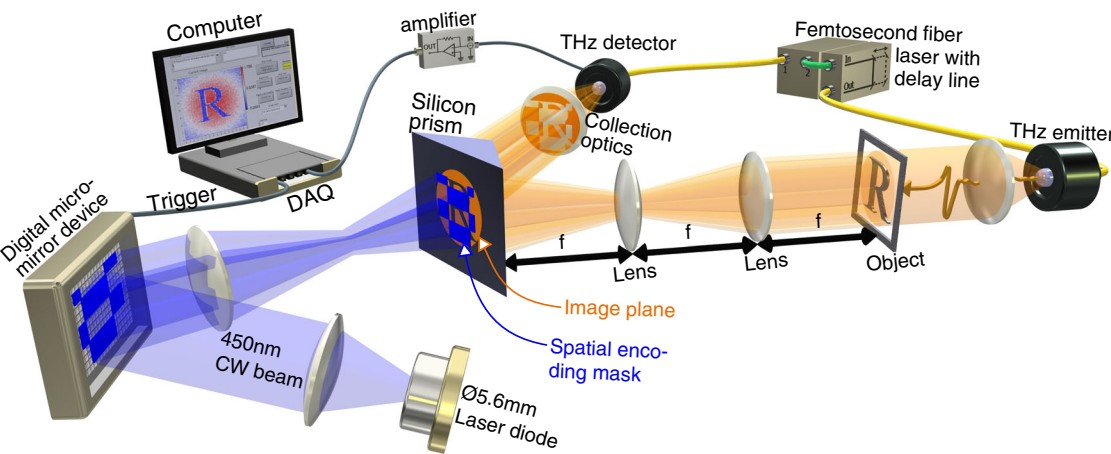

**Fig. 2 Experimental illustration.** A collimated THz pulse is launched at an object, an image of the object is projected onto the top interface of a silicon prism with the THz incident angle being 30°. Simultaneously, this top interface has a 2D conductive pattern imprinted onto to it via the photoexcitation of a 450 nm continuous-wave light beam, which itself has been spatially patterned by a DMD, thereby imparting a 2D encoding mask onto the THz beam. The reflected THz is collected onto a single-pixel photoconductive THz antenna and detected signals for many different encoding masks are recorded. Note, many of the visible-light projection optics have been left out for visual clarity.

**Software optimization**. The software part of the system needs to model the temporal response of the physical components and significantly improve the overall SNR preferably without needing long post-processing times. Least-squares fitting algorithms are very fast and they can be used to fit our measured signals to the equations describing the physics of our system. This allows us to adequately account for the component or process with the slowest temporal response associated with it, whether it is the detector response time, the modulator switch rate or the amplification electronics bandwidth for example. In our case, it is switching the silicon's material response from dielectric to conductor via the photoexcitation of electron–hole pairs, with all other processes and components ignored as they were instantaneous compared to the response of the photocarrier lifetimes. The carrier concentration is described by $N(t) = Be^{-t/\tau} \int_{-\infty}^{t} G(t')e^{t'/\tau} \mathrm{d}t'$ for carrier lifetime $\tau$, carrier generation rate $G(t)$ and arbitrary amplitude constant $B$. If we say that we switch on our light at $t = 0$ and turn it off at time $t = T$, then we get

$$N(t) = \begin{cases} B(1 - e^{-t/\tau}) & \text{for } 0 \leq t \leq T \\ B(1 - e^{-T/\tau})e^{-(t-T)/\tau} & \text{for } t > T \end{cases}. \quad (3)$$

Knowledge of this needs to be combined when processing Eq. (1) in order to reconstruct our image. The usual processing of Eq. (1) requires the length of $Y$ to equal the number of rows in $A$, namely one measurement for each projected spatial mask. Given that we change the masks at regular periods $T$ then the temporal response for each mask will be given by Eq. (3). Therefore, we can fit this equation to our measurement using $B$ as the fitting parameter, as we know $\tau$, obtaining an estimate for the mask amplitude-value at $t \gg \tau$. This estimated value can then be inputted into Eq. (1) for image reconstruction. For this, we need to find all the $B$ values that describe the temporal-segments of each individual mask (see Supplementary Note 3 for fast calculation methodologies). The calculation of these values takes into account the system's time-response, therefore compared with an averaging or integrating processing approach this method will give a better estimate when the masks have been switched at a rate comparable to $\tau$.

In Fig. 3a, we show the signals measured with our DAQ card and the reconstructed least-squares fits to those signals for mask switch rates of 2, 5, 10 and 20 kHz, noting that the optical delay line was kept fixed at the max of the THz field cycle. The silicon used had a lifetime of $\tau = 250$ μs resulting in a switch rate limit of

around 3–5 kHz, thus only the 2 kHz switch rate reached a near-stable value as can be seen in Fig. 3a. To test our algorithm's ability to predict the final-equilibrium value while measuring on timescales shorter than $\tau$, we compare the reconstructed images using the values from our algorithm and averaging the signals over each mask duration. We image two metallic lines placed in the middle of a Gaussian THz-beam profile. The resulting signal-to-noise ratio (SNR), defined as the mean value of the transmissive areas divided by the standard deviation of the opaque areas (Supplementary Fig. 2), is shown in Fig. 3b as a function of mask switch rate. The least-squares fitting algorithm and the averaging approach both start off with similar SNRs, however as the switch rate increases the averaging approach experiences a much sharper decrease in SNR compared to the least-squares algorithm. Fitting $S = aR^{-b}$ to the SNR, $S$, and switch rate, $R$, reveals $a = 40(60)$ and $b = 0.91(0.44)$ for the averaging (least-squares) approaches. In other words, if the mask projection periods are on timescales comparable to the system's temporal response, taking the mean signal-value during each mask periods results in a linear drop in SNR with switch rate. Whereas mathematically accounting for the temporal behavior results in a square-root decrease. In the end, our proposed technique demonstrates that we can obtain an image at a 20 kHz mask modulation rate, only limited by the DMD switch-speeds, whereas averaging approaches are limited to a maximum modulation rate of 6–8 kHz due the physical processes in our system. However, it needs to be mentioned that this is only meant to demonstrate how to use post-processing to overcome switch rate limitations imposed by physical phenomena as the actual signals used were an average of 16 measurements to remove the various noise sources in our system. Applying a carrier-wave modulation to the emitter and modifying the equations can be used to reduce noise, as shown in Supplementary Note 5.

**Undersampling**. Undersampling techniques are often used in single-pixel imaging modalities to reduce the acquisition time. Compressed sensing[35] is a technique used to reduce the number of measurements needed to image an object, i.e. undersample the object. It works by directly measuring a sparse representation of an object and trying to recover the image with optimization algorithms, which typically require computational times that prevent real-time display of the images. Fortunately, quick-image reconstruction techniques from undersampled data do exist.

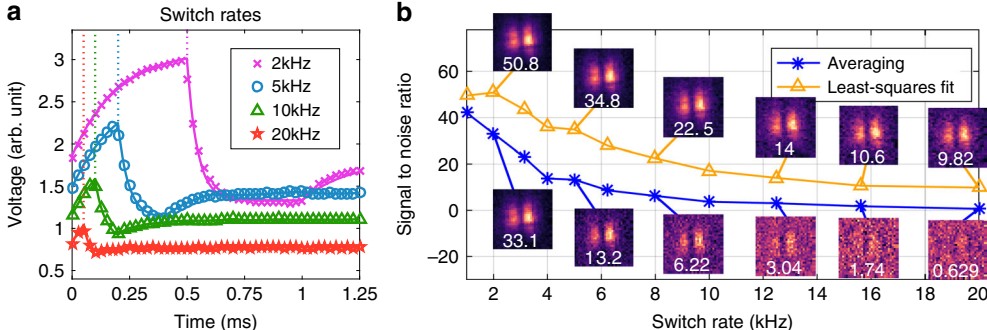

**Fig. 3 Signal with switch rate. a** The measured voltage against time for different mask switch rates. The markers are the raw data and the solid lines are the fitted signals. The vertical dotted lines indicate when the first mask was changed for the different switch rates. Note that the data has been vertically offset for visual clarity. **b** Image signal-to-noise ratio (SNR) against mask switch rate where we have used the proposed least-squares fitting algorithm (yellow-triangles) and average over the mask duration (blue-asterisks) to estimate the mask values used in image reconstruction. See Supplementary Note 4 for definition and calculation of SNR. Images for 2, 5, 8, 12.5, 15.625 and 20 kHz switch rates are shown with their SNR in white. Note, the signals were acquired by fixing the delay line at the maximum of the electric field cycle.

Namely, partially measuring the Fourier spectrum of an object by projecting sinusoidal stripes of known spatial frequency, and then obtaining the image via inverse Fourier Transform[22,24,42]. Alternatively, one can calculate a pseudoinverse matrix $A^{-1}$, store it in memory and obtain the image via fast matrix multiplication $\Psi = A^{-1}Y$[43].

The Fourier transform undersampling technique[22,24,42] has the benefit that most images of interest are sparse in the spatial frequency domain, thus accurate image recovery is obtained from a small number of Fourier coefficients. However, measuring the coefficients requires the projection of sinusoidal stripes, in other words masks with greyscale values. DMDs are inherently limited to being either on or off, 1s and 0s, where the projection of greyscale values is obtained by temporal dithering at the expense of reducing the mask switch rate[24,44]. Spatial dithering to achieve values between 1 and 0 introduces quantization errors[42,44], thus this method is not optimal when implemented with DMDs.

Pre-calculating a reconstruction matrix and obtaining the image via fast matrix multiplication allows free selection of the projected masks. However, simply calculating the pseudoinverse creates undersampling artefacts and lower stability with noise. Fortunately, Czajkowski et al.[43] developed a theory that allows for the calculation of a pseudoinverse matrix that also includes a set of spatial filters that regularize the image, resulting in images of similar quality to the slow $L_1$ minimization algorithms[43].

THz modulators are still in their infancy of being developed prior to commercialization. Here, as demonstrated in "Modulation geometry" section, THz modulation is achieved via the generation of charge carriers in silicon via the photoexcitation of silicon and a DMD is used to spatially pattern the optical pump beam. Therefore we need to use binary masks to achieve the quickest acquisition rates. Masks derived from the Hadamard transform have been shown to have higher noise robustness compared to random masks in the THz regime[33], and are inherently binary. Their noise robustness arises as they are orthogonal and have the property of minimizing the mean squared error in all the image pixels[25]. Although this noise robustness property is also shared with the Fourier masks, we chose the Hadamard basis due to its binary nature.

Hadamard matrices have values of either $+1$ or $-1$ with the property that the rows are orthogonal to one another. Usually one uses the Sylvester construction to create a Hadamard matrix, which yields masks of different spatial frequencies as shown in Supplementary Note 6. It is then necessary not to omit important low spatial-frequency masks, which requires prior knowledge of

the imaging scene. Fortunately, alternative construction techniques of Hadamard matrices exist; namely the Paley construction method. Here a cyclic construction is used where each row is obtained by shifting the row above it[45]. This results in masks that are simply shifted from one another as shown in Supplementary Fig. 4. Therefore, all these masks have similar spatial frequencies, hence no need for special selection of the important masks.

We use masks constructed from a Paley type-I Hadamard matrix which has values of 1s and $-1$s, whereas we can only project 1s and 0s. Adding a secondary differential measurement for each mask allows the correct encoding to be obtained, as shown in ref. [46], which does increase the SNR at the expense of doubling the number of measurements[33]. In Fig. 4, we compare the performance of [1 0] and [1 $-1$] masks by plotting the similarity structure index (SSIM) as we increase the sampling to image-pixels ratio. Looking at the [1 0] masks, we see that the SSIM begins to rise with some oscillations and then saturates at a value around 0.7 once the sampling ratio becomes above 40%. The reason for this saturation is that adding more measurements becomes unnecessary to construct a better image, however noise in the measurements prevent the SSIM from improving. The differential [1 $-1$] masks reach a higher SSIM in the end, however the undersampling oscillations occur until a sampling ratio of 80% is reached. The increase in SNR at the end is due to the differential measurement having superior noise robustness, however doubling the measurements means the undersampling artifacts last for twice as long. These results show that the use of [1 0] masks with 40% sampling ratio is the optimum for our system.

**Application**. The culmination of the ideas in this work, namely the modulation geometry, software processing and under-sampling, pave the way for a rapid THz spectroscopic imaging system without incurring great cost or complexity. In the Supplementary Video 1, we show the real-time THz image reconstructed as we move a metal line around the field-of-view (the delay line was fixed at the peak of the THz field). The displayed image has a resolution of $32 \times 32$ pixels, and the achieved number of frames per second is around 6. The mask switch rate was 4 kHz, meaning with a 40% sampling ratio we should expect around 10 frames per second; however, for each image there is ~50 ms of waiting time needed to synchronize the DMD, DAQ card and the THz delay line. This waiting time can be eradicated with a more integrated system design.

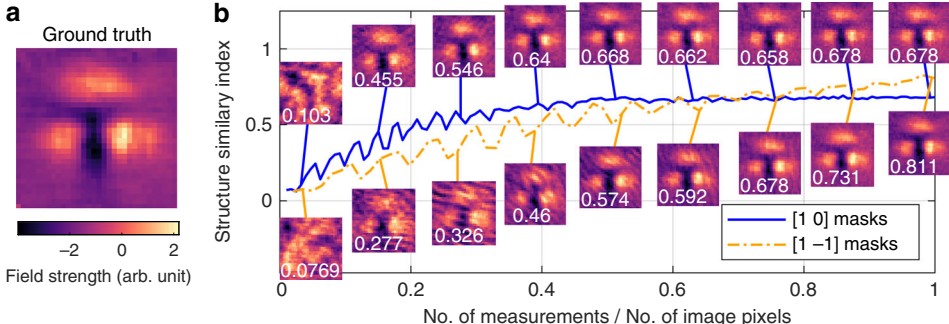

**Fig. 4 Hadamard Undersampling. a** The image used as the ground truth for undersampling investigations; an average of 9 images taken consecutively. **b** Structure similarity index (SSIM) as a function compression ratio (number of mask measurements/number of image pixels). Solid-blue (dash dotted-yellow) line is data from [1 0] ([1 −1]) Paley–Hadamard masks (see Supplementary Fig. 4 for mask examples). Some images at various compression ratios are shown with a solid line indicating the sampling ratio of the image. Note, the optical delay line was fixed at the peak of the THz field cycle.

To ensure that our approach does not obstruct the ability of measuring spectroscopic THz images, in Supplementary Note 7 we image an object consisting of four materials: air, metal, 210 µm and 420-µm-thick plastic. For spectroscopic imaging, however, the full temporal waveform at each image pixel needs to be recorded, or equivalently images need to be acquired at every temporal point of the THz-pulse cycle. This results in 3D data, two *xy*-dimensions and one temporal dimension, which is shown in the Supplementary Video 2 (time of the image is shown as the vertical dashed line on the THz pulse below the image). From these data, the obtained transmission and phase delay values are what we expect as shown in the Supplementary Note 7. Therefore, this confirms that the undersampling and data processing do not have any negative effects on spectroscopic THz imaging. See ref. [5] for a basic overview on THz data analysis. Note that this does increase the total acquisition time due to the necessity of recording images at many temporal points, with the increase being linearly proportional to the number of time points needed to be sampled. This unfortunately makes the acquisition of a spectroscopic THz image in real-time more difficult, however temporal undersampling of the THz waveform[47] reduces this problem.

To demonstrate an application of this technique we track hydration changes in a leaf, from an *Achyranthes aspera* plant, as it is illuminated by an intense white-light source without access to water. As the illumination is too strong for the leaf, it experiences high oxidative stress resulting in a loss of chlorophyll[48]. The loss of chlorophyll after the experiment can clearly be seen in Fig. 5a. In Fig. 5b–h (i–o), we show the THz transmission (phase delay) images as the leaf experiences photo-damage. In the first transmission and phase delay images, Fig. 5b, i, we can see the main leaf vein, which appears as white/green color in the transmission and the phase delay images, corresponding to low transmission and large phase delay. The reason is that the leaf veins are thicker than the rest of the leaf and also transport water, thus the increased absorption and phase delay. Then, as the leaf becomes photo-damaged, we can see the phase delay decrease and the transmission increase in the photo-damaged area. This is because the damaged area becomes thinner under photo-oxidative damage, which results in lower phase delay and increased transmission. The thinning of the leaf is because under intense optical illumination, the chloroplasts inside the cell collapse reducing the overall plant size[49]. Comparing the transmission and phase images, we see that the phase images give clearer contrast of the veins. This suggests that there is higher water content near the veins whereas the leaf thickness remains the same. Finally, the phase delay inside the vein also decreases over time. This is because the leaf is running out of water causing

the leaf to shrink[50]. This shows we are able to track very subtle hydration changes in vivo.

## Discussion

This work optimizes single-pixel THz cameras, where a photo-conductive THz emitter and detector pair was used, and the spatial THz modulator shines spatially patterned visible light on silicon. The spatial control of THz is of critical importance, thus we show that a total internal reflection modulation geometry achieves the lowest insertion losses and highest modulation depth. This relaxes the modulator parameters, which allows for a reduction in total cost without reducing the efficiency. Every system has a temporal response, which can be exploited to our advantage. In our system, the photoexcitation of silicon with a CW source has the slowest response, hence we decrease noise by fitting our measurements to the relevant photocarrier equations. With the fitting approach, we are able to demonstrate successful imaging with a modulation speed of 20 kHz, only limited by the DMD switch-speeds, compared to 8 kHz for an averaging approach. Next, we further improve acquisition speeds by undersampling, however the typical compressed sensing algorithms involve complex minimization algorithms preventing real-time image display. For this reason, we use a partially sampled Hadamard matrix along with a regularized image reconstruction matrix to reduce computational times, finding a 40% sampling ratio using [1 0] Hadamard masks to be optimum here. This work executes alternative ideas to the previously demonstrated single-pixel THz imaging works achieving 32 × 32 resolution with ~6 frames per second, greatly outperforming the previous state-of-the-art of 8 × 8 resolution at 1 frame per second[26]. We demonstrate a possible biological application where we are able to track photo-induced damage in a leaf as well as subtle hydration changes in the non-damaged regions in vivo.

Future work will focus on obtaining sub-wavelength resolution by limiting carrier diffusion via the creation of micro-sized silicon pillars and putting the object in the near-field regime. Simultaneously, there will be an effort to reduce noise by employing a secondary photodiode to measure the fiber-laser power fluctuations. Further increases in acquisition speeds will be obtained by exploiting the temporal sparsity of THz pulses for further undersampling as well as achieving more efficient synchronization between the system's components. This works opens up the possibility of rapid THz-spectroscopy imaging systems at commercially viable prices without any complex manufacturing or by having to sacrifice the spectroscopic capacities of THz–TDS systems. Further, the work presented here could be used to design no-moving parts imaging systems that are based on the self-interference properties of THz quantum cascade lasers, such as

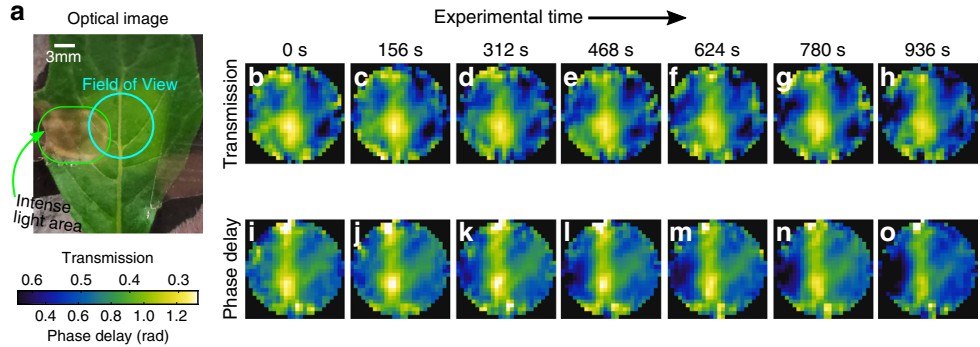

**Fig. 5 THz leaf imaging. a** Optical image of the leaf, from an *Achyranthes aspera* plant, used in the experiment. The THz field-of-view is indicated here along with the area illuminated by an intense white-light source (~3 sun illumination). **b–h** (**i–o**) THz transmission (phase delay) of the leaf shown in part **a** at different points in time (experimental time shown on top). Note that the transmission images show an average from 0.3 to 0.8 THz, $T = \sum_{i=1}^{n} |E_{\text{samp}}(\omega_i)/E_{\text{ref}}(\omega_i)|/n$, and the phase delay images show the averaged unwrapped phase, $\Phi = \sum_{i=1}^{n} \text{unwrap}(\arg(E_{\text{samp}}(\omega_i)/E_{\text{ref}}(\omega_i)))/n$, with the transmission colourscale been inverted. Further, the THz images have been deconvolved, using *deconvblind* function in Matlab R2016a, by calculating the point-spread function of our imaging system by Fourier optics.

those in reference[29], by replacing the linear *xy*-stage with a spatial light modulator.

## Data availability

The raw experimental data, in MATLAB data format, and relevant MATLAB processing scripts needed to reproduce Fig. 4 can be obtained on Figshare https://doi.org/10.6084/m9.figshare.11955327. The data and scripts for any other results shown in this work can be obtained by contacting the corresponding author.

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

## Acknowledgements

This work was partially supported by the Research Grants Council of Hong Kong (Project Numbers 14206717, 14201415 and AoE/M-05/12), The Hong Kong Innovation and Technology Fund (Project Number ITS/371/16), The Engineering and Physical Sciences Research Council (Grant Number EP/S021442/1), National Natural Science Foundation of China (project number 51777023) and the Royal Society Wolfson Merit Award (E.P.M.). We thank Xudong Liu for providing suitable silicon samples and helpful discussions along with Edward P. J. Parrot.

## Author contributions

R.I.S., T.B. and E.P.-M. conceived the idea. R.I.S. created the experimental setup and programmed all equipment. T.B. designed the least-squares fitting algorithm. X.Y. designed the undersampling procedures. R.I.S. performed the experiments and data analysis. R.I.S. wrote the manuscript and all other authors provided editorial input.

## Competing interests

The authors declare no competing interests.
