## [Peer Review File · Nature Communications]

Reviewers' comments:

Reviewer #2 (Remarks to the Author):

Review of manuscript entitled: Real-time THz imaging with a single-pixel detector, by Stantchev et al., submitted to Nature Communications

I have carefully read the answers of the authors to the previous round of review. On this basis, I do not see any comparison to more powerful imaging techniques, already demonstrated (e.g., -Yamagiwa, M. et al. Journal of Infrared, Millimeter, and Terahertz Waves 39, 561, (2018), "Real-Time Amplitude and Phase Imaging of Optically Opaque Objects by Combining Full-Field Off-Axis Terahertz Digital Holography with Angular Spectrum Reconstruction" - Locatelli et al. Scientific Reports 5, 13566 (2015) "Real-time terahertz digital holography with a quantum cascade laser")

That make use of digital holography. This comparison is necessary also in view of the remarks of one of the reviewers (and of the consequent addition to the text by the authors) that points out how microbolometer arrays are broadband and room temperature detectors that can be easily deployed in higher performance set-ups, like digital holography ones, that also for this reason must be properly compared and cited.

Other concerns are:

I have difficulties to understand the spectroscopic application at the end of the article. They use Fourier Transform spectroscopy to get the different hyperspectral images of plastic layers and metal. Therefore they use a delay line I don't see in the setup description. The time needed to browse the entire temporal impulse can definitely limit the possibility to achieve real-time spectroscopic imaging, to my mind.

In view of the comments above, I think the authors must convincingly provide the appropriate answers before the manuscript can be considered for publication

Reviewer #3 (Remarks to the Author):

I thank the reviewers for addressing the points I mentioned in my previous review. In my opinion, the manuscript can be published now.

Responses to reviewers of NCOMMS-19-38842-T

Reviewer #2 (Remarks to the Author):

Review of manuscript entitled: Real-time THz imaging with a single-pixel detector, by Stantchev et al., submitted to Nature Communications

I have carefully read the answers of the authors to the previous round of review. On this basis, I do not see any comparison to more powerful imaging techniques, already demonstrated (e.g.,

-Yamagiwa, M.et al. Journal of Infrared, Millimeter, and Terahertz Waves 39, 561, (2018), "Real-Time Amplitude and Phase Imaging of Optically Opaque Objects by Combining Full-Field Off-Axis Terahertz Digital Holography with Angular Spectrum Reconstruction"

- Locatelli et al. Scientific Reports 5, 13566 (2015) "Real-time terahertz digital holography with a quantum cascade laser")

That make use of digital holography. This comparison is necessary also in view of the remarks of one of the reviewers (and of the consequent addition to the text by the authors) that points out how microbolometer arrays are broadband and room temperature detectors that can be easily deployed in higher performance set-ups,

like digital holography ones, that also for this reason must be properly compared and cited.

We thank the reviewer for pointing out these two references which we had overlooked, and they have now been added as references 17 and 18. They indeed are powerful THz imaging techniques as they have measured a THz amplitude and phase image in real-time, however this is for a single THz frequency. Whilst microbolometer arrays do detect THz radiation over a broadband range, they do not allow the different frequencies to be separated. Thus a spectral image with such a setup would require one to sweep the frequency of the emitter and record an image at each frequency, thereby creating a problem similar to us having to measure the full THz waveform using a delay line (your concern below). Further, such detectors do not have the ability to measure sub-picosecond phenomena like THz time-domain spectrometers. This is fine for many current industrial applications, however as technology based on ultra-fast dynamics matures this will become a much more limiting feature.

We therefore have added the following sentences to the introduction on line 27:

“In fact, microbolometer arrays have been used to obtain amplitude and phase images at a single THz frequency via the use of digital holography [17, 18]. However, spectral images would require taking multiple images at different frequencies and would not achieve the sub-picosecond temporal resolution granted by THz-spectrometers, a capacity needed for observing ultra-fast dynamics.”

Other concerns are:

I have difficulties to understand the spectroscopic application at the end of the article. They use Fourier Transform spectroscopy to get the different hyperspectral images of plastic layers and metal. Therefore they use a delay line I don't see in the setup description.

Sorry for missing that information out initially. For our experimental setup we modified a commercial THz time-domain spectrometer, the MENLO K15, and the optical delay line came together with the fibre laser. This has now been clarified by modifying the relevant label in figure 2. We thank the reviewer for pointing out this essential detail. Furthermore, on line 127 we have added the following paragraph:

“Note, in this work we use a commercial THz time-domain spectrometer (MENLO K15), which uses a pair of fibre coupled photo-conductive antennae as the emitter-receiver pair. An optical delay line changes the temporal arrival times of the femtosecond pulses incident onto the emitter and receiver allowing the measurement of the temporal THz waveform. A DAQ card (NI-USB 6351) reads out the detector and bias state of the emitter instead of the built-in Lock-In amplifier.”

The time needed to browse the entire temporal impulse can definitely limit the possibility to achieve real-time spectroscopic imaging, to my mind.

This is true and provides motivation for approaches such as those in this paper and others to work around this limitation. We have added a statement to explicitly point this out on line 420:

“This unfortunately makes the acquisition of a spectroscopic THz image in real-time more difficult, however temporal undersampling of the THz waveform [47] reduces this problem.”

In view of the comments above, I think the authors must convincingly provide the appropriate answers before the manuscript can be considered for publication.

We hope the reviewer finds the above changes are satisfactory.

Reviewer #3 (Remarks to the Author):

I thank the reviewers for addressing the points I mentioned in my previous review. In my opinion, the manuscript can be published now.

We thank the reviewer for his time and effort in improving our manuscript. His contributions in the previous review phase have certainly improved this work.

REVIEWERS' COMMENTS:

Reviewer #2 (Remarks to the Author):

The Authors have addressed all my concerns and I can now endorse publication in Nature Communications.